# CryoEM structure of the low-complexity domain of hnRNPA2 and its conversion to pathogenic amyloid

Jiahui Lu[1,2], Qin Cao [1,2], Michael P. Hughes[1,2,3], Michael R. Sawaya [1,2], David R. Boyer [1,2], Duilio Cascio [1,2] & David S. Eisenberg [1,2✉]

hnRNPA2 is a human ribonucleoprotein (RNP) involved in RNA metabolism. It forms fibrils both under cellular stress and in mutated form in neurodegenerative conditions. Previous work established that the C-terminal low-complexity domain (LCD) of hnRNPA2 fibrillizes under stress, and missense mutations in this domain are found in the disease multisystem proteinopathy (MSP). However, little is known at the atomic level about the hnRNPA2 LCD structure that is involved in those processes and how disease mutations cause structural change. Here we present the cryo-electron microscopy (cryoEM) structure of the hnRNPA2 LCD fibril core and demonstrate its capability to form a reversible hydrogel in vitro containing amyloid-like fibrils. Whereas these fibrils, like pathogenic amyloid, are formed from protein chains stacked into β-sheets by backbone hydrogen bonds, they display distinct structural differences: the chains are kinked, enabling non-covalent cross-linking of fibrils and disfavoring formation of pathogenic steric zippers. Both reversibility and energetic calculations suggest these fibrils are less stable than pathogenic amyloid. Moreover, the crystal structure of the disease-mutation-containing segment (D290V) of hnRNPA2 suggests that the replacement fundamentally alters the fibril structure to a more stable energetic state. These findings illuminate how molecular interactions promote protein fibril networks and how mutation can transform fibril structure from functional to a pathogenic form.

[1] Departments of Chemistry and Biochemistry and Biological Chemistry, University of California, Los Angeles, Los Angeles, CA, USA. [2] UCLA-DOE Institute, Molecular Biology Institute, Howard Hughes Medical Institute, Los Angeles, CA, USA. [3] Present address: Department of Cell and Molecular Biology, St. Jude Children's Research Hospital, Memphis, TN, USA. ✉email: david@mbi.ucla.edu

The protein hnRNPA2 functions in RNA stabilization, splicing, trafficking, and translation[1–4]. In its role in protecting mRNAs, it is a component of cytoplasmic stress granules, one of the so-called membraneless organelles (MLOs)[1,5]. Its 341 amino-acid-residue sequence contains an RNA-binding domain (RBD) with two RNA recognition motifs (RRMs) and a 161-residue C-terminal low-complexity domain (LCD). The LCD is rich in Gly, Tyr, Phe, Asn, and Asp residues, and poor in hydrophobes, with Val, Ala, Leu, and Ile entirely absent. Regions of biased amino acid composition, such as this, allow transient interactions between the many known MLO-forming proteins[1,5–8].

Aggregated hnRNPA2 has been found to have nuclear clearance and cytoplasmic inclusions under cellular stress[2], and in vitro its LCD forms fibrils with the capacity to interact with each other to make hydrogels of the sort observed in MLOs[1,6]. In vitro studies have shown evidence of liquid droplets over time turning into hydrogels containing fibrils[9,10]. The hnRNPA2-LCD hydrogel can trap other functional proteins such as hnRNPA1 and CIRBP[1], suggesting a functional role for the hydrogel.

The fibrils of MLOs are functional in contrast to pathogenic fibrils associated with neurodegenerative conditions. One such condition is the disease MultiSystem Proteinopathy (MSP) with symptoms indistinguishable from ALS and FTD. Patients with this disease present with aggregated hnRNPA2 in cytoplasmic inclusions[2] and the variant sequence D290V in the LCD of hnRNPA2. Other mutations found in the RNP LCDs were previously shown to impede MLO and hydrogel formation[1,11,12].

Informing both functional MLO-associated fibrils and pathogenic disease-associated fibrils, hnRNPA2 is similar to other RNA-binding proteins, including hnRNPA1[13], FUS[7,9,12,14], TIA1[11], and TDP-43[15,16]. For this reason, knowledge of the structures of these proteins, and of their LCDs in particular, may uncover general principles of how proteins can form both functional and pathogenic fibrils. Near-atomic structures are already available for the ordered fibril cores of the LCDs of FUS[14] and TDP-43[15].

Here we determined the cryoEM structure of the LCD of hnRNPA2 and the crystal structure of the segment containing the variant sequence D290V. These structures help to answer the questions: (1) What sequence features account for the formation of the functional fibrils? (2) What structural features of the functional fibrils drive them to bind non-covalently with each other to form the networks that underlie hydrogels? And (3) How does a single residue mutation convert functional to pathogenic fibrils?

## Results

**Hydrogel and fibril formation of mC-hnRNPA2-LCD.** Purified recombinant mC-hnRNPA2-LCD was concentrated to ~60 mg ml$^{-1}$ and incubated at 4 °C to test its ability to form a hydrogel. Determination of whether the sample exists in a liquid or gel phase is assayed by the mobility of a bubble through the sample. We inverted a 1.5 ml silicon tube containing freshly purified mCherry-tagged (purple color) hnRNPA2-LCD (termed mC-hnRNPA2-LCD), and then introduced a bubble at the lower surface. The bubble rose to the top of the tube, indicating a homogenous solution rather than a gel (Fig. 1a, left). Within a week, the protein solution converted into a gel, indicated by the retention of bubbles at the lower surface (Fig. 1a, right). Negative-stain transmission electron microscopy (TEM) of the diluted hydrogel showed a network of uniform amyloid-like fibrils with an average width of 20 nm (Fig. 1b). The X-ray diffraction pattern is consistent with cross-β architecture: two reflections at 4.7 and 10 Å corresponding to the inter-strand and inter-sheet spacing, respectively[17], of amyloid fibrils (Fig. 1c).

**Reversibility of mC-hnRNPA2-LCD hydrogel.** To test whether the hnRNPA2-LCD hydrogel is reversible, we performed a reversibility assay on purified mC-hnRNPA2-LCD incubated at 4 °C for either 2 days or 2 weeks. After heating the 2-day hydrogel to 45 °C, the bubbles introduced stayed at the bottom of the sample, indicating maintenance of a hydrogel state (Supplementary Fig. 1a, left). TEM images showed networks of uniform, twisted, and unbranched amyloid-like fibrils similar to those observed before heating (Supplementary Fig. 1b, left). However, after heating to 55 °C, introduced bubbles slowly rose and stalled in the middle of the hydrogel (Supplementary Fig. 1a, middle). TEM of these samples showed fragmented fibrils and partial disruption of fibril networks (Supplementary Fig. 1b, middle). After heating to 65 °C and above, the hydrogel transitioned to a homogenous protein solution where bubbles introduced quickly rose to the top of the silicon tube (Supplementary Fig. 1a, right), indicating the 2-day hydrogel is reversible. TEM revealed that over 80% of the fibrils are dissolved, leaving behind amorphous aggregates (Supplementary Fig. 1b, right). For the 2-week hydrogel, bubbles introduced remained immobile up to 75 °C heating (Supplementary Fig. 1c) and TEM images (Supplementary Fig. 1d) revealed a fibril morphology similar to the 2-day hydrogel. Hence, the 2-week hydrogel appears less reversible than the 2-day hydrogel. Moreover, the 2-week fibrils at 65 °C were more bundled than the remnant of 2-day hydrogel fibrils at 65 °C. We infer that the young hnRNPA2 fibrils are mostly reversible, but aging makes the fibrils irreversible, possibly by the evolution of fibril bundling. We also purified and concentrated the mCherry tag alone as a negative control, and we found that fibril networks and the amorphous aggregates were not the result of mCherry itself (Supplementary Fig. 2).

**CryoEM structure of the mC-hnRNPA2-LCD fibrils.** To answer the question of why the mC-hnRNPA2-LCD hydrogel exhibits reversibility, we determined the cryoEM structure of mC-hnRNPA2-LCD fibrils to a resolution of 3.1 Å (Fig. 2c). During cryoEM data collection and processing, we found that the mC-hnRNPA2-LCD fibrils have only a single morphology, which consists of one asymmetric protofilament with a pitch of 600 Å, a left-handed helical twist of −2.88° and a helical rise of 4.81 Å (Fig. 2b and Supplementary Fig. 3). Contrary to the globular proteins which form 3D structures, the mC-hnRNPA2-LCD structure is confined to 2D layers which stack on top of each other, forming a twisted in-register β-sheet that runs along the fibril axis. Out of 161 residues in the LCD, 57 residues from Gly263 to Tyr319 form a fibril core (Fig. 2a). The mC-hnRNPA2-LCD fibril encompasses a PY-nuclear localization signal (PY-NLS) and the core hexamer segment containing the site for a disease-causing mutation that was identified by Kim et al. in 2013[2] (Fig. 2d). Previous studies have shown that mutations in the PY-NLS can lead to ALS[18]. An aromatic triad composed of two Tyr (Y275 and Y283) residues and one Phe (F309) residue is buried in the center of the fibril, stabilizing the structure through π–π interactions (Supplementary Fig. 4). Data collection and refinement statistics are summarized in Table 1.

**Energy analysis explains hnRNPA2-LCD fibrils' reversibility.** To quantify the structural features that led to the reversibility of hnRNPA2-LCD fibrils, we calculated the solvation energy for our mC-hnRNPA2-LCD fibril structure (details in "Methods" section). We obtained the solvation energy per layer of the fibril and an averaged solvation energy per residue. We illustrated these energies in a solvation energy map where each residue is colored according to its energy. The solvation energy for mC-hnRNPA2-LCD fibril core (−19.5 kcal mol$^{-1}$ per chain and −0.34 kcal mol$^{-1}$

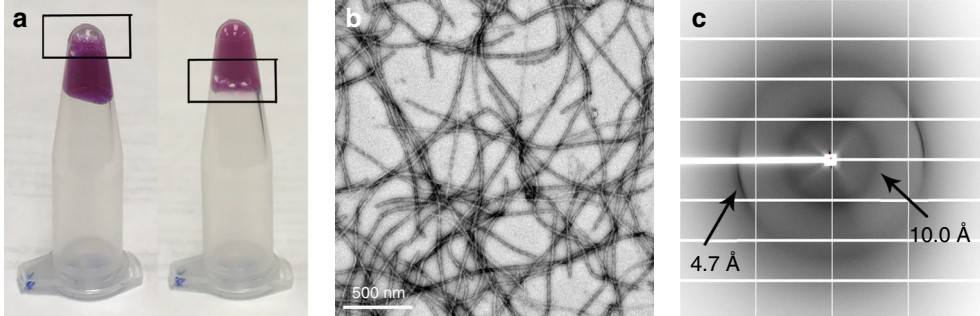

**Fig. 1 mC-hnRNPA2-LCD forms a reversible hydrogel that contains amyloid-like fibrils displaying the cross-β diffraction pattern. a** Left micro-centrifuge tube holds a homogenous solution of freshly purified mC-hnRNPA2-LCD. A bubble (box) rises to the top of the tube. The right tube contains the hydrogel formed by concentrated mC-hnRNPA2-LCD; the bubble is trapped by the hydrogel. **b** Hydrogel formed by mC-hnRNPA2-LCD visualized by transmission electron microscopy shows uniform, amyloid-like fibrils. Scale bar: 500 nm. **c** X-ray diffraction pattern of the hydrogel formed by mC-hnRNPA2-LCD. The two reflections characteristic of amyloid are visible at 4.7 and 10 Å spacings.

**Fig. 2 The Cryo-EM structure of the mC-hnRNPA2-LCD fibril core. a** Domain structure of full-length hnRNPA2. The LCD (residues 181–341) is identified for structural determination. The gray bar shows the range of the ordered fibril core of the cryoEM structure. The red bar shows the core segment (crystal structure described below) containing the site of a disease-causing mutation. The magenta bar shows the nuclear localization signal, PY-NLS. The sequence of the ordered region is shown below with corresponding colors. **b** The mC-hnRNPA2-LCD fibril reconstruction, showing its left-handed twist and pitch. **c** Density and atomic model of one cross-sectional layer of the fibril. The box shows the aromatic triad. **d** Atomic model of one cross-sectional layer of the fibril. The predicted LARKS domain (contains 7 LARKS motifs) (above) and the shorter LARKS motif (below) are colored orange; the core segment is colored cyan, with its disease-causing mutation site colored red; the nuclear localization signal is colored magenta (lower left). β-sheet-forming residues are G274-N277, Y288-D290, P303-S306, S312-N314, indicated by arrows.

per residue) is comparable to another well-known reversible protein fibril, FUS ($-12.2$ kcal mol$^{-1}$ per chain and $-0.20$ kcal mol$^{-1}$ per residue) (Fig. 3a, b). In contrast the solvation energy of pathogenic irreversible amyloid structures such as human serum amyloid A[19] ($-34.4$ kcal mol$^{-1}$ per chain and $-0.64$ kcal mol$^{-1}$ per residue) has a more negative value per chain and residue (Fig. 3c), indicating that compared to the pathogenic amyloid fibrils, mC-hnRNPA2-LCD fibrils form less stable structures. Solvation energies of other proteins are compared in Table 2.

**The disease-causing mutant core segment**. Muscle biopsies have found atrophic hnRNPA2 fibrils in patients diagnosed with inclusion body myositis (IBM)[2]. A missense point mutation that converts a conserved aspartic acid to a valine (D290V) in hnRNPA2-LCD was identified in 2013 as being associated with ALS and MSP[2]. To investigate the structural differences between the wildtype and the mutant, we crystallized the hexamer segment ²⁸⁶GNYNVF²⁹¹ (Supplementary Fig. 5) containing the disease-causing mutation and determined its structure by X-ray crystallography (Fig. 4a and Statistics in Table 3). As expected, the crystal structure of the mutant segment forms a steric zipper motif and the sidechains of the two β-sheets mate tightly with each other in a dry interface. The area buried (Ab) and shape complementarity (Sc) for the mutant segment are 104 Å² per chain and 0.86, respectively. The mutant segment structure forms in-register, parallel β-sheets. To compare this mutant segment crystal structure with our cryoEM wildtype mC-hnRNPA2-LCD structure, we overlaid the two structures (Fig. 4b). The backbones of the wildtype structure and the mutant segment fit well with each other, whereas the sidechain of Asp290 in the wildtype structure clashes with Asn289 of the mating sheet of mutant steric zipper, disrupting the steric zipper formation. On the other hand, forming the steric zipper in the mutant structure causes Asn287 and Tyr288 sidechains to adopt different conformations, which create steric clashes with nearby Asn282 and Met304 residues in the wildtype fibril structure. Moreover, Phe291 from the mating sheet of the steric zipper clashes with the backbone of Gly281 in the wildtype fibril structure. Based on this structural analysis, we speculate that the D290V mutation converts the reversible hnRNPA2 fibrils to irreversible fibrils. When the conserved Asp290 residue is mutated to a Val, the mutation enables a more stable steric zipper interaction that was previously prevented by the Asp290 sidechain. Meanwhile, the formation of a steric zipper causes multiple steric clashes and disrupts the wildtype fibril structure, shifting the aggregation of hnRNPA2 from reversible to irreversible.

**Table 1 Statistics of cryoEM data collection, refinement, and validation.**

|  | HnRNPA2-LCD (EMD-21871) (PDB: 6WQK) |
|---|---|
| *Data collection and processing* |  |
| Magnification | ×130,000 |
| Voltage (kV) | 300 |
| Electron exposure (e⁻ Å⁻²) | 34.5 |
| Defocus range (μm) | 0.8–5.1 |
| Pixel size (Å) | 1.064 |
| Symmetry imposed | C1; Helical |
| Initial particle images (no.) | 529,821 |
| Final particle images (no.) | 132,571 |
| Map resolution (Å) | 3.1 |
| FSC threshold | 0.143 |
| Map resolution range (Å) | 200–3.1 |
| *Refinement* |  |
| Initial model used | De Novo |
| Model resolution (Å) | 3.2 |
| FSC threshold | 0.5 |
| Model resolution range (Å) | 200–3.2 |
| Map sharpening *B* factor (Å²) | −120 |
| *Model composition* |  |
| Non-hydrogen atoms | 2070 |
| Protein residues | 285 |
| *B factors (Å²)* |  |
| Protein | 55.3 |
| *R.m.s. deviations* |  |
| Bond lengths (Å) | 0.006 |
| Bond angles (°) | 0.89 |
| *Validation* |  |
| MolProbity score | 1.70 |
| Clashscore | 5.59 |
| Poor rotamers (%) | 0.00 |
| *Ramachandran plot* |  |
| Favored (%) | 94 |
| Allowed (%) | 6 |
| Disallowed (%) | 0.00 |

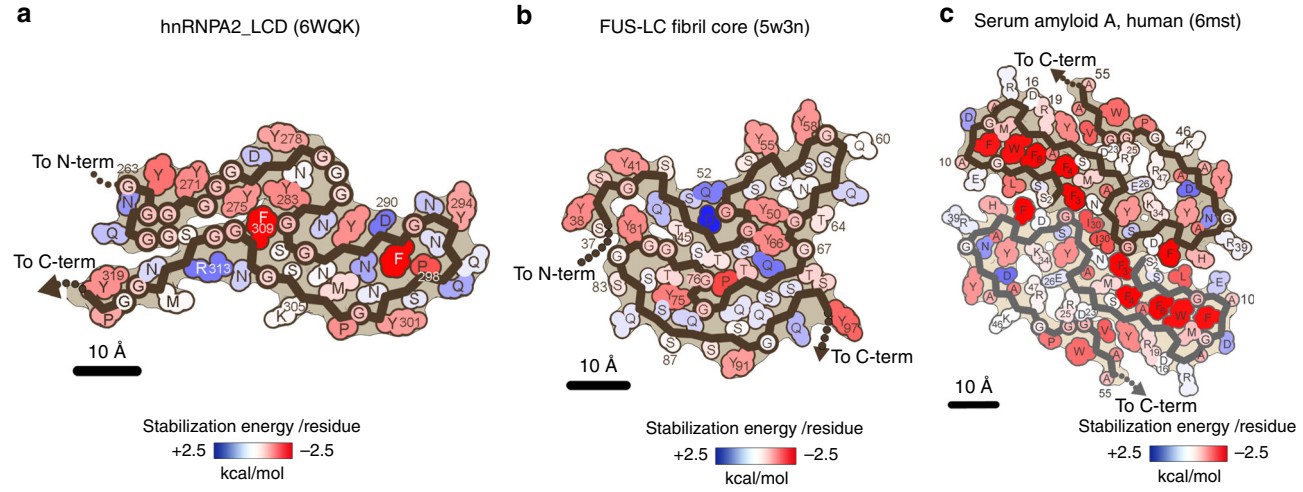

**Fig. 3 Solvation energy maps of mC-hnRNPA2-LCD, FUS-LCD, and serum Amyloid A.** Residues are colored according to their stabilization energies from unfavorable (blue, +2.5 kcal mol$^{-1}$) to favorable (red, −2.5 kcal mol$^{-1}$). **a** Solvation energy map of mC-hnRNPA2-LCD ordered fibril core. **b** Solvation energy map of FUS-LCD[14] ordered region. **c** Solvation energy map of human serum amyloid A[19], ordered region.

**Table 2 Comparative values of solvation standard free energies of functional and pathogenic fibrils.**

| | Ordered residues | Method | Resolution (Å) | Energy/chain (kcal mol$^{-1}$) | Energy/residue (kcal mol$^{-1}$) |
|---|---|---|---|---|---|
| hnRNPA2-LCD (functional) | 57 | CryoEM | 3.1 | −19.5 | −0.34 |
| FUS-LCD (functional) | 61 | ssNMR | 2.7 | −12.2 | −0.20 |
| Serum amyloid A Human (pathogenic) | 54 | CryoEM | 2.7 | −34.4 | −0.64 |
| Tau PHFs AD patient (pathogenic) | 73 | CryoEM | 3.4 | −28.9 | −0.40 |
| Aβ42 (pathogenic) | 42 | CryoEM | 4.0 | −24.8 | −0.59 |

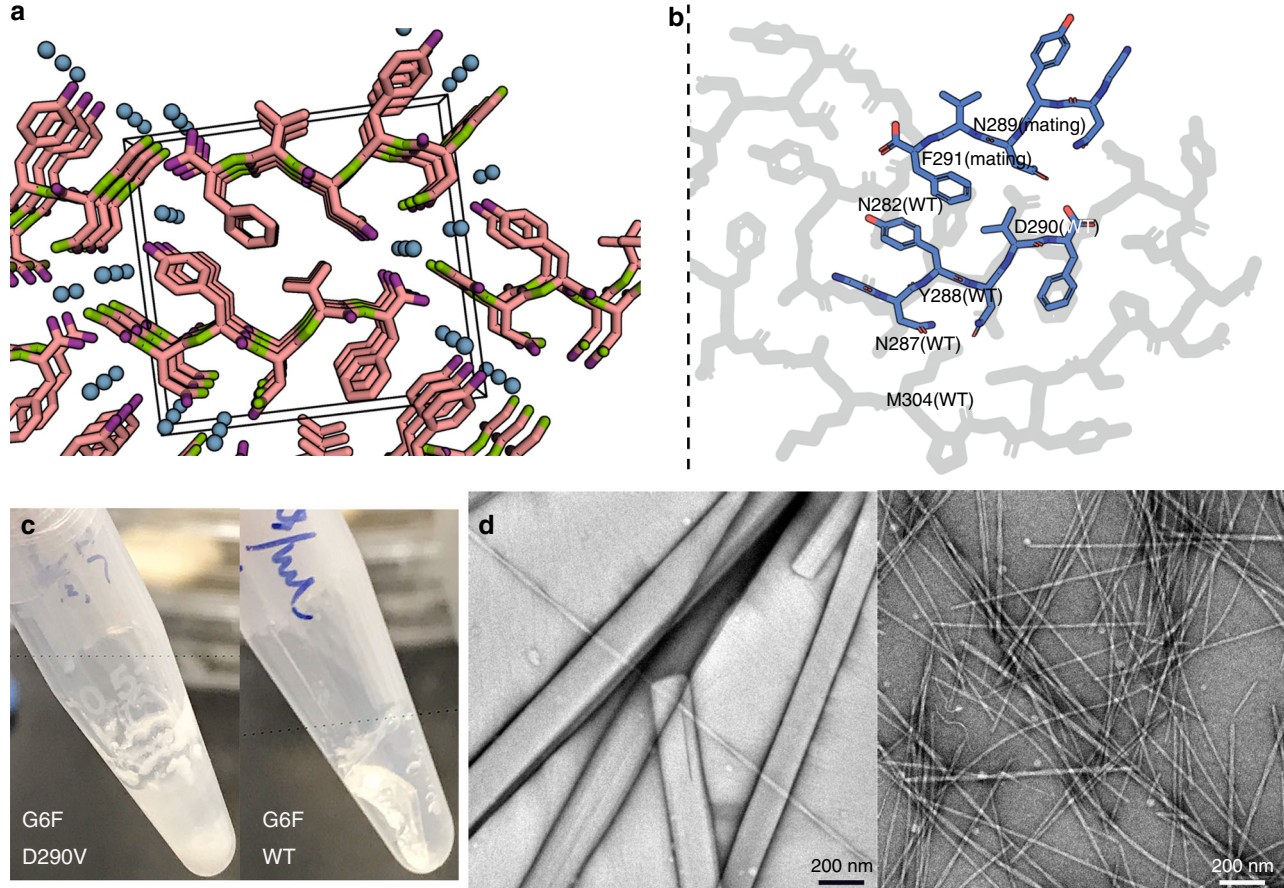

**Fig. 4 Mutant core segment and the corresponding wildtype segment show distinct features. a** The steric zipper structure determined for mutant segment GNYNVF, viewed down the fibril axis. Water molecules within the structures are shown as aqua spheres. The unit cell is shown as a box. **b** A superimposition of the mutant steric zipper structure of GNYNVF (blue) on the cryoEM mC-hnRNPA2-LCD structure (gray) shows the incompatibility of the mutant structure with the fold of the wildtype fibril. Only the corresponding part of the wildtype structure is shown. The sidechains of Asn287, Tyr288, and Asn289 of the mutant structure experience steric clashes with the sidechains of Asn282, Asp290, and Met304 of the wildtype structure. **c** Left: Mutant segment GNYNVF shaken after 4 days forms a white precipitate. Right: Wildtype segment GNYNDF shaken after 4 days forms a hydrogel. **d** Left: The mutant segment GNYNVF (from **c**) visualized by transmission electron microscopy, shows wide needle crystals. Scale bar: 200 nm; Right: Wildtype segment GNYNDF (from **c**) visualized by transmission electron microscopy, shows thin fibrils.

To compare the effects on aggregate formation of the wildtype core segment GNYN**D**F and the mutant segment GNYN**V**F, both were shaken for 4 days and the samples were checked using negative stain TEM (details in "Methods" section). The mutant segment, as we expected, precipitated after shaking (Fig. 4c, left), and TEM images showed that the precipitates were composed of micro-crystals (Fig. 4d, left). The wildtype segment forms a hydrogel (Fig. 4c, right), which to our knowledge is the shortest segment that forms a hydrogel, and TEM images indicated that the gel was composed of thin fibrils (Fig. 4d, right). The results suggest that the mutant segment with the pathogenic mutation forms a more stable structure than the WT segment, and converts a gel-promoting segment into an aggregation-promoting segment.

## Discussion

mC-hnRNPA2 LCD fibrils reveal distinctly different structural and energetic properties from pathogenic amyloid (Table 4). Our TEM and X-ray diffraction data show that mC-hnRNPA2-LCD forms fibril networks with cross-β structures (Fig. 1). Unlike pathogenic amyloid fibrils, which are usually not crosslinked and are stable at high temperatures[15,16,20], mC-hnRNPA2-LCD fibrils form 3-dimensional networks and are sensitive to even mild heat

(Supplementary Fig. 1). Heated mC-hnRNPA2-LCD, instead of remaining as a fibril network, is broken into oligomer-like material, and does not reform a hydrogel upon cooling. We hypothesize that in this in vitro experiment, the environment lacks biological factors such as chaperones to achieve reversibility. Nevertheless, the smaller stabilization solvation energy compared to pathogenic amyloid indicates the reversibility of hnRNPA2 LCD fibrils (Fig. 3). Amyloid-like fibrils have also been reported for other granule-associated, LCD-containing proteins, such as FUS[1,21] and hnRNPA1[13], those being heat-sensitive and SDS-sensitive, and being capable of forming a reversible hydrogel.

Our cryoEM structure of mC-hnRNPA2-LCD fibrils reveals local structural motifs that distinguish reversible from pathogenic amyloid: mC-hnRNPA2-LCD fibrils contain multiple weakly binding LARKS motifs[22] and only a single short steric zipper motif[23]. LARKS are low-complexity amyloid-like, reversible kinked segments which enable amyloid-like assembly with reduced stability compared to steric zippers[22]. Steric zippers are pairs of β-sheets in which the sidechains of one β-sheet inter-digitate with those of the opposing sheet to form a tight, dry interface[23,24]. Unlike steric zippers, the β-sheets formed by LARKS are often kinked either at a Gly or aromatic residues[22], limiting the size of interacting surfaces. Consequently, LARKS are

less stabilizing than steric zippers. Within the mC-hnRNPA2-LCD fibril core there are eight predicted LARKS (Hughes et al., in preparation), seven grouped together (Supplementary Fig. 6a and Fig. 2d), each corresponding to a kink in the protein backbone. The structure contains only four short β-sheets, reducing the possibility of steric zippers (Fig. 2d). The multiple LARKS motifs introduce kinks into the backbone of the structure, disrupting stabilizing β-sheets into segments shorter than the six residues of most steric zippers, thus contributing to the reversibility of the hnRNPA2-LCD fibrils. Pathogenic amyloid typically houses a high content of β-sheets capable of forming steric zippers, for example Tau PHFs[25] (78%) and Aβ42[26] (74%) in contrast to reversible amyloid-like fibrils hnRNPA2 (23%) and FUS[14] (26%). We note the location of the short β-sheets in hnRNPA2-LCD are consistent with the secondary structure prediction of Murray et al.[27] (Fig. 2d). We regard LARKS and steric zippers as the structural basis of "stickers"[28,29] in reversible, amyloid-like fibrils and pathogenic amyloid fibrils, enabling inter-protein and intra-protein interactions.

A further feature that distinguishes reversible amyloid-like fibrils from irreversible, pathogenic fibrils is the enrichment of polar residues in the fibril core. As shown in the mC-hnRNPA2-LCD residue polarity map, almost 50% of the residues are hydrophilic and only 12% are hydrophobic (Supplementary Fig. 6b), and the core contains a high content of Gly (35%), Asn (18%), and Tyr (16%) compared to tau PHFs[25] (Gly:12%, Asn:4%, Tyr:1.4%) and Aβ42[26] (Gly:14%, Asn:2%, Tyr:2%). Furthermore, hnRNPA2-LCD lacks any instance of the hydro-phobic residues Val, Ala, Ile, and Leu. In contrast, tau PHFs[25] and Aβ42[26] contain 24% and 36%, respectively, of these residue types (Table 5). The high content of Gly contributes to hnRNPA2-LCD flexibility. 3D environment profiling[30] also suggests a higher percentage of residues exposed to solvent (35%) and a lower content of nonpolar residues (4%) compared to tau PHFs[25] (12% and 13%) and Aβ42[26] (14% and 18%) (Table 6). These differences are also manifested in solvation energy maps (Fig. 3). Our results indicate that hnRNPA2-LCD adopts a cross-β fibril assembly that morphologically resembles pathogenic amyloid. However, quan-titative examination shows that reversible amyloid-like fibrils differ from pathogenic amyloid by having a smaller solvation stabilization energy and a greater enrichment of LARKS and polar residues. These features facilitate hydrogen-bonding with water molecules and contribute to LCD flexibility and capacity for assembly and disassembly in stress granules.

HnRNPA2 and FUS[14], both proteins that function in stress granules, form fibrils which share four structural features. First, both are mainly stabilized by hydrogen-bonding and polar interactions rather than hydrophobic effects. Both hnRNPA2-LCD and FUS-LCD fibril cores are enriched in Tyr residues that stabilize structures by π-stacking interactions[14,24,31]. Similar to hnRNPA2-LCD, the FUS-LCD fibril core has ample polar

**Table 3 Statistics of X-ray crystallography data collection and refinement.**

|  | GNYNVF PDB: 6WPQ |
|---|---|
| *Data collection* |  |
| Space group | P2$_1$ |
| *Cell dimensions* |  |
| *a, b, c* (Å) | 4.78, 19.00, 20.74 |
| *α, β, γ* (°) | 90.00, 95.71, 90.00 |
| Resolution (Å) | 1.1 |
| *R*$_{merge}$ (%) | 11.6 (21.8) |
| *I /σI* | 10.4 (5.7) |
| Completeness (%) | 86.5 (41.7) |
| Redundancy | 5.6 (3.5) |
| *Refinement* |  |
| Resolution (Å) | 20.64–1.10 |
| No. of reflections | 1196 |
| *R*$_{work}$/*R*$_{free}$ (%) | 7.5/10.7 |
| *No. of atoms* |  |
| Protein | 51 |
| Water | 3 |
| *B-factors* (Å$^2$) |  |
| Protein | 4.8 |
| Water | 7.4 |
| *R.m.s. deviations* |  |
| Bond lengths (Å) | 0.014 |
| Bond angles (°) | 1.804 |

**Table 4 Comparative properties of functional amyloid-like fibrils and pathogenic amyloid fibrils.**

|  | Functional amyloid-like fibrils | Pathogenic amyloid fibrils |
|---|---|---|
| Variety | Monomorphic | Tend to be polymorphic |
| Stability | Tend to be labile and reversible | Tend to be stable and irreversible |
| ΔG° of stabilization | ~ −0.2 kcal mol-of-residue$^{-1}$ | ~ −0.4 kcal mol-of-residue$^{-1}$ |
| Protein chains stack into β-sheets by backbone H-bonds | Yes | Yes |
| Sheets pair | Yes, by LARKS | Yes, by steric zippers |
| Residues that drive sheet pairing | Often Asn, Gln, Tyr, Phe, Ser, Gly, Pro in low-complexity domains | Varied, but frequently alternating polar and apolar especially Val, Ile, Val |
| Fiber diffraction pattern | Cross-β | Cross-β |
| Morphology | Crosslinked networks of fibrils | Non-crosslinked deposits of fibrils |

residues (Table 5, columns 2 and 3), and FUS-LCD has a high content of glutamine residues that are either exposed to solvent or form stabilizing hydrogen-bonded ladders along the fibril axis. The abundance of residues with hydroxyl groups in both hnRNPA2 and FUS (Table 5, column 3) allows post-translational modifications, such as phosphorylation[32], to modulate the stability of the fibrils, affecting stress granule assembly. Compared to hnRNPA2-LCD, the FUS-LCD has a relatively high content of Ser and Thr residues, allowing additional stabilization of the core structure through sidechain hydrogen bonding.

Second, solvation energy calculations suggest that the stability of the mC-hnRNPA2-LCD fibril structure is nearly as poor as FUS-LCD. Because the LCDs in hnRNPA2 and FUS are reversible fibrils associated with stress granules, we expect both to have lower stability than pathogenic amyloid fibrils. And this is confirmed by our solvation energy calculation (Fig. 3b and Table 2). The relatively small solvation stabilization energy explains why hnRNPA2 and FUS are sensitive to mild heat and denaturing condition such as 2% SDS[16].

Third, hnRNPA2 and FUS fibrils exhibit only a single morphology. In contrast, a majority of the studied structures of pathogenic irreversible amyloid have shown polymorphism, whether the fibrils were produced ex vivo or in vitro. For example, so far, cryoEM and ssNMR studies have identified nine morphologies for different isoforms of pathogenic protein tau[25,33], three morphologies for TDP-43[15], and six morphologies for Aβ42[26,34–38]. Moreover, multiple fibril polymorphs have been structurally determined under a single set of conditions from the same EM grid[15,39]. Unlike the polymorphism seen in the pathogenic irreversible fibril structures, our cryoEM structure of mC-hnRNPA2-LCD fibril shows only a single morphology under the condition tested. This lack of polymorphism is also seen in the FUS-LCD structure that was determined by ssNMR[14]. As suggested previously, monomorphic structure is essential for proper biological functions of proteins[14]. Similar to globular proteins, reversible amyloid-like fibrils have apparently been honed by evolution to adopt a single structure to support a given function. Reversible fibrils such as hnRNPA2 can undergo folding and unfolding to find a kinetically accessible global free energy minimum, leading to an optimal structure of lowest accessible free energy. The hnRNPA2-LCD is conserved among distantly related organisms (Supplementary Fig. 7), further evidence of its

functional role. Fourth, both FUS-LCD and mC-hnRNPA2-LCD fibrils are composed of a single protofilament rather than two or more, as is the case with more than 70% of pathological fibrils structurally determined.

Enrichment in LARKS motifs also diminishes the tendency for polymorphism. Interactions between two LARKS motifs are more specific than steric zippers since complementarity between kinked surfaces requires a particular sequence, whereas complementarity between two canonical flat β-sheets can be achieved by more sequences. Furthermore, a kinked sheet of a given sequence has few plausible ways to fit with a partner kinked sheet, whereas two flat β-sheets have a range of plausible registrations with respect to each other to achieve different low-energy structures. The polymorphism seen in pathogenic irreversible fibrils represents different structures that are trapped in multiple local free energy minimum. The hnRNPA2 being reversible and functional in vivo implies that it should also follow the Anfinsen's dogma[40] that the native full-length hnRNPA2 fibrils in vivo should have the same structure as the recombinant protein structure we report here.

Our cryoEM hnRNPA2-LCD structure is consistent with previous biochemical and NMR studies. Xiang et al. in 2015[6] reported that the hnRNPA2-LCD adopts a similar structure in hydrogels and in liquid droplets as evidenced by conservation of NAI footprints which probed 23 residues of hnRNPA2-LCD from Ser219 to Ser335. They found the N-terminal portion was unprotected from NAI labeling, but the C-terminal portion encompassing N282–Y324 was protected. This protected region overlaps well with our ordered core spanning residues G263–Y319. The identity of the only unprotected probe residue in this range, Lys305, agrees with our cryoEM structure showing that the sidechain of Lys305 points toward the solvent. There is also agreement between the partial protection of residues Asn282, Tyr288, and Ser312 and our observation that all three are partially buried in the fibril core. Similar consistency is found between the strong protection of Ser285, Ser306, and Ty319 and our observation that these residues are completely buried (Fig. 2c). The structural consistency among different structural studies of hnRNPA2-LCD, specifically our cryoEM, Murray et al.'s NMR[27], and Xiang et al.'s NAI footprinting[6], suggests two important points: (1) the fibril core structure of hnRNPA2-LCD is identical or closely similar among three studies, (2) the interactions of residue sidechains we report are closely similar as Xiang et al.'s observations in both hydrogel and liquid droplets.

We offer a speculative hypothesis of how a single-point missense mutation can alter the fibril structure. This hypothesis is consistent with the data we have. As indicated by its greater stabilization energy, the mutant segment GNYNVF is more prone to form crystals than the wild type segment GNYNDF (Fig. 4). In addition, we calculated the stabilization energy of the mutant segment and all hexamer segments that constitute the wild type hnRNPA2 LCD fibril, FUS LCD fibril, and pathogenic amyloids. Unsurprisingly, the mutant segment is an outlier in terms of having greater energetic stability than any hexameric segment in either hnRNPA2 or FUS (Supplementary Fig. 8a). This energetic analysis supports our hypothesis that when this mutation forms, the mutant segment can drive the assembly of a more stable fibril.

If the wild type segment GNYNDF of full-length hnRNPA2 assembles as it does in the peptide crystal structure, the remaining

**Table 5 Comparative sequence composition of functional and pathogenic amyloid fibril cores.**

| | Gly (%) | Asn+Gln (%) | Tyr+Ser+Thr (%) | Val+Ala+Ile+Leu (%) |
|---|---|---|---|---|
| HnRNPA2-LCD (functional) | 35.1 | 22.8 | 22.8 | 0.0 |
| FUS-LCD (functional) | 19.7 | 18.0 | 59.1 | 0.0 |
| Tau PHFs (pathogenic) | 12.3 | 8.2 | 15.1 | 24.4 |
| Aβ42 (pathogenic) | 14.3 | 4.8 | 7.2 | 35.7 |

**Table 6 Comparative 3D–1D environmental profiles of hnRNPA2-LCD with pathogenic amyloids.**

| Protein | Exposed to solvent (%) | Partially buried with high fractional environmental polarity (%) | Nonpolar buried (%) |
|---|---|---|---|
| hnRNPA2-LCD | 35 | 32 | 4 |
| Tau PHFs | 12 | 30 | 13 |
| Aβ42 | 14 | 24 | 18 |

residues would necessarily adopt a different conformation than the wild type fibril in order to avoid steric clash with the hexapeptide assembly. Our cryoEM images offer evidence that the mC-hnRNPA2-D290V-LCD fibrils are more heterogeneous and thicker than the wild type mC-hnRNPA2-LCD fibrils (Supplementary Fig. 8b, c). This supports our hypothesis that the D290V mutation drives the mutant segment GNYNVF to adopt a homozipper conformation and drives the fibril assembly into a different structure with two protofilaments.

Previous studies by Guo et al.[41] showed that over half of the LCD-containing RNPs such as FUS, TAF15, hnRNPA1, and hnRNPA2 possess a PY-NLS. They found that the import receptor karyopherin-β2 (Kapβ2) binds the PYNLS signal of these RNPs and then, once bound, acts as chaperones that dissolve preformed fibrils. The PYNLS of hnRNPA2 (residues 296–319), containing two epitopes: a hydrophobic/basic residue stretch (reside 196–312) and a consensus sequence R/K/H-X$_{(2-5)}$-PY (residue 313–316), is located right after a central steric zipper motif (segment $^{286}$GNYNDF$^{291}$) that drives hnRNPA2 fibrilization[2]. Our structure of hnRNPA2 provides insight into how this interaction of Kapβ2 with hnRNPA2 functions to prevent fibrilization from occurring.

The hnRNPA2 fibril core is essentially a highly kinked, branched β-arch. The potential fibril-driving steric zipper motif (286–291) forms the only extended (unkinked) β-strand in the fibril structure and contributes to the upper half of the β-arch (Fig. 2d). The lower half of the β-arch is part of the PYNLS signal (residues 296–319). The PYNLS signal forms a hydrophobic pocket between P298 and N300 that buries the terminal aromatic residue of the steric zipper F291 in the fibril core, yielding a significant amount of the stabilization energy (Fig. 3a). We infer that if Kapβ2 binds to the PYNLS signal, the bottom half of the β-arch is unavailable to make the hydrophobic environment necessary to sequester F291 to stabilize the central steric zipper and form fibrils. In addition, we propose that owing to the labile nature of the hnRNPA2 fibrils, monomers may more easily dissociate from the fibril and be sequestered by Kapβ2. This is supported by the relatively weak stabilization energy of hnRNPA2 fibrils compared to irreversible amyloid as the energy barrier between monomers associated with the fibril end or freely floating is much lower. The mechanism of action that our structure suggests is that hnRNPA2 fibrils cannot form when associated with the Kapβ2 via the PYNLS[41] because the PYNLS is required to support the steric zipper motif that drives fibril formation[2].

Our cryoEM mC-hnRNPA2-LCD structure is consistent with and presents more evidence to support Guo et al.'s[41] discovery that the PYNLS is included in the structure yet not completely buried in the middle of the fibril core (Fig. 2c, d), which could allow chaperones such as Kapβ2[42] to engage and melt the hnRNPA2-LCD hydrogel.

Does the mCherry-tag affect the structure of mC-hnRNPA2-LCD fibrils? We actively sought to characterize the hnRNPA2 LCD fibril structure without the mCherry tag; however, we found that hnRNPA2 LCD alone is highly insoluble and tends to aggregate. We thus included the mCherry tag in our experiments. mCherry is visible during cryoEM processing. As shown by EM images, the width of the mC-hnRNPA2-LCD fibril (~20 nm) (Supplementary Fig. 9a) is significantly larger than that of our hnRNPA2-LCD fibril core model (2.6–6.7 nm) (Supplementary Fig. 9d, e). During cryoEM data processing, we found globular densities surrounding the fibrils in low-resolution 2D classifications (Supplementary Fig. 9b), and these globular densities were averaged into a fuzzy coat in high-resolution 2D classifications and 3D reconstruction (Supplementary Figs. 3c and 9d). The width of the fibril including the globular densities or fuzzy coat in 2D classification is about 22 nm which matches the width of the

fibrils measured in EM micrographs. The average diameter of each globular density is about 2.9 nm and is compatible with the diameter of the crystal structure of mCherry protein[43] (2.8 nm, PDB 2H5Q), which suggests these globular densities correspond to mCherry (Supplementary Fig. 9c), as shown by the in-scale fitting of our model of mC-hnRNPA2-LCD fibril and the crystal structure of mCherry to the averaged 2D class image (Supplementary Fig. 9c). There are three reasons that we consider innocuous the effect of such a large tag on the core structure. First, there are 86 amino acids between the mCherry-tag and the start of the fibril core (N-terminal G263), including a four-residue linker and the beginning segment of the LCD. We think that an 86-residue-long flexible peptide allows enough freedom to accommodate the mCherry tags without distorting the fibril core structure. When the fluorescent tag is sufficiently separated from the fibril core, the tag can freely fold around the fibril core as a "fuzzy coat", so that fibrillization and core packing are not disturbed[44]. Second, the biological, full-length hnRNPA2 contains two globular N-terminal RRMs totaling 193 residues. Although the mCherry-tag is a little larger than the RRMs, it is possible that the mCherry tag provides a mimic of the natural RRM domains in the full-length fibril. Protein–RNA interactions are important for assembling RNP granules in that RNA molecules act as scaffolds for multivalent RNA-binding proteins[45] such as hnRNPA2. These globular densities of mCherry suggest a potential arrangement of the RRMs as they wrap around the hnRNPA2-LCD fibril in vivo. With enough sequence length between the RRMs and the fibril core-forming region, the RRMs could interact with RNA molecules to facilitate the RNP granule assemblies[46]. Third, the mCherry tag "fuzzy coat" is not visible in the cryoEM 3D classification (Supplementary Fig. 9d), suggesting there are no disruptive interactions between the large mCherry tag and the hnRNPA2 LCD fibril core. Thus, the mCherry tag may suggest a model for fibril function.

Our studies indicate that hnRNPA2 fibrils together with other RNPs such as FUS share intrinsic structural and energetic properties that distinguish them from pathogenic amyloid in two major ways: (1) more LARKS motifs and fewer steric zippers, (2) enrichment of polar residues and scarcity in hydrophobic residues. These functional amyloid-like fibrils are likely the constituents of liquid condensates that contain proteins with LCDs. Experiments to detect the cross-β diffraction signal from fibrils of hnRNPA2 in solution are ongoing and made challenging due to background scattering arising from solvent and the β-sheet-rich mCherry tag (Supplementary Fig. 9). High protein concentration or mutations can drive such functional fibrils into more stable pathogenic amyloid fibrils. Cells need to maintain a delicate thermodynamic balance: biomolecules need to be concentrated enough to overcome the entropic loss associated with assembly into cellular condensates. Yet condensates can be easily converted by mutations, crossing a high kinetic barrier[47] to form irreversible fibrils in MLOs that feed forward into disease states.

## Methods

**Materials and purification of mCh-hnRNPA2-LCD fusion protein**. The construct for overexpression of mCherry-hnRNPA2-LCD fusion protein was provided by Dr. Masato Kato of University of Texas, Southwestern, and has the sequence shown in Supplementary Fig. 10. Protein overexpression and purification procedure was adapted from the protocol by Kato et al. (2012)[1]. Protein was overexpressed in *E. coli* BL21(DE3) cells with 0.5 mM IPTG at 25 °C for overnight. LB media with 0.1 mg ml$^{-1}$ ampicillin was used for cell culture. Harvested cells were resuspended in lysis buffer containing 50 mM Tris–HCl pH 8.0, 500 mM NaCl, 2 M guanidine HCl, 2 mM TCEP, and Halt$^{TM}$ protease inhibitor cocktail (Thermo Scientific) for 30 min on ice, and then sonicated. The cell lysate was centrifuged at 32,000×g for an hour. The supernatant was filtered and loaded onto a HisTrap HP column(GE healthcare) for purification. The HisTrap column was pre-equilibrated with the lysis buffer. After proteins were loaded onto the column, proteins were washed with the lysis buffer until the UV280 spectrum line became flat. The sample then

was washed with a gradient from 100% wash buffer containing 25 mM Tris–HCl pH 8.0, 150 mM NaCl, 2 M Guanidine HCl, 20 mM imidazole, and 2 mM TCEP to 100% elution buffer containing 25 mM Tris–HCl pH 8.0, 150 mM NaCl, 2 M Guanidine HCl, 300 mM imidazole, and 2 mM TCEP. Eluted proteins were flash frozen by liquid nitrogen and stored at −80 °C for future use.

**Formation of mCherry-hnRNPA2-LCD hydrogels**. Purified mCherry-hnRNPA2-LCD fusion proteins were dialyzed overnight at room temperature against a dialyzing buffer containing 20 mM Tris–HCl pH 7.5, 200 mM NaCl, 20 mM BME, 0.5 mM EDTA, and 0.1 mM PMSF. The protein solutions were concentrated to 40–80 mg ml$^{-1}$. The protein solutions (~100 μl) were filled into tightly sealed 1.5 ml silicon tubes, and the tubes were incubated at 4 °C for 1–3 days.

**Negative stain TEM and X-ray diffraction**. All protein samples for TEM were diluted 10 times using dialysis buffer. Samples for TEM were prepared by applying 4 μl of sample on glow-discharged 400 mesh carbon-coated formvar support films mounted on copper grids (Ted Pella, Inc.). The samples were allowed to adhere for 2 min and 30 s, and washed twice with water. The samples were then stained for 2 min with 2% uranyl acetate and allowed to dry for 1 min. Each grid was inspected using a T12(FEI) electron microscope.

For fiber diffraction, the procedure followed the protocol described by Rodriguez et al.[48]. The hydrogel sample placed between two capillary glass rods, and allowed to air dry. When the fibrils are completely dry, the glass rods with fibrils aligned in between were mounted on a brass pin for x-ray diffraction. For homogenous solution, 10 μl of 60 mg ml$^{-1}$ protein solution is pipetted into a PET sleeve and then mounted on a brass pin. Buffer was used as a control and used for background subtraction. X-ray diffraction data were collected at beamline 24-ID-E of the Advanced Photon Source, Argonne National Laboratory, Argonne, IL, USA, at a wavelength of 0.979 Å and temperature of 100 K. Data were collected using 2.5° oscillations and 450 mm detector distance with an EIGER detector. The results were analyzed using the Adxv software[49].

The 3D classification of mCherry-hnRNPA2-LCD structure indicates the helical rise of the fibrils is 4.8 Å. This is different from the results we got from the hydrogel fiber diffraction, showing the inter-strand spacing to be 4.7 Å. We speculated that this difference is due to the sample preparation. For fiber diffraction, fibrils are dried between the two glass rods at room temperature for hours before being shot with X-ray. Yet CryoEM samples are dried by being blotted for only several seconds. Fibrils being drier might lead to a denser packing of the structures. Therefore in in vitro experiments, attentions should be given to the sample preparation steps.

**Reversibility assay of mCherry-hnRNPA2-LCD hydrogels**. Bubbles were introduced to the hydrogels as a way to tell if the hydrogels are reversible. Hydrogels were heated up from 4 to 75 °C with a 10 °C increment, at each temperature samples were incubated for 10 min in a PCR machine. 2 μl of samples at each temperature were taken and diluted 10 times with the dialysis buffer, and checked by negative stain EM.

**CryoEM data collection, reconstruction, and model building**. 2.5 μl of diluted hydrogel samples at a concentration of 0.43 mg ml$^{-1}$ were applied to glow-discharged Quantifoil Cu R1.2/1.3, 300 mesh carbon grids. Samples were blotted with filter paper to remove excess sample and then plunge-frozen in liquid ethane using a Vitrobot Mark IV (FEI). Cryo-EM data were collected on a Gatan K2 Summit direct electron detector on a Titan Krios (FEI) microscope equipped with a Gatan Quantum LS/K2 Summit direct electron detection camera (operated with 300 kV acceleration voltage and slit width of 20 eV). Super-resolution movies were acquired with a pixel size of 1.07 Å pixel$^{-1}$ (0.535 Å pixel$^{-1}$ in super-resolution movie frame). Thirty movie frames were recorded each with a frame rate of 5 Hz using a dose rate of 1.15 e$^-$ Å$^{-2}$ frame$^{-1}$ for a final dose of 34.5 e$^-$ Å$^{-2}$ on the sample. Automated data collection was driven by the Leginon automation software package[50]. 2935 micrographs were collected with a defocus range from 0.8 to 5.1 μm.

Micrographs summing all frames were all corrected for gain reference and then micrographs with a group of three frames were used to estimate the contrast transfer function using CTFFIND 4.1.8[51]. Unblur[52] was then used to correct beam-induced motion with dose weighting and anisotropic magnification correction, leading to a physical pixel size of 1.064 Å pixel$^{-1}$. Micrographs with crystalline ice, severe astigmatism, or obvious drift were discarded. All subsequent data processing were performed using RELION 2.0[53,54].

All filaments were picked manually using EMAN2 e2helixboxer.py[55]. Particles were first extracted using a box size of 1024 and 686 pixels with an inter-box distance of 10% of the box length. 2D classification using 1024 pixel particles was used to estimate the fibril pitch and helical parameters. We also performed 2D classifications with 686 pixel particles to select particles for future 3D classification. We performed Class3D jobs with three classes and manually controlled the tau_fudge factor and healpix_order to obtain resolutions around 6–7 Å, using an elongated Gaussian blob as an initial reference. We selected particles that contribute to the highest resolution class and generated an initial 3D reconstruction by running Class3D with 1 class. To obtain a higher resolution reconstruction, we re-extracted particles with a box size of 224 pixels from the fibril tubes containing the Class3D-selected 686 pixel particles. All 224 pixel particles were used directly for two rounds of 3D classification. The final subset of selected particles was used for high-resolution gold-standard refinement as described previously[54]. The final overall resolution estimate was evaluated to be 3.1 Å based on the 0.143 FSC cutoff[56]. Projections from the final reconstruction closely match the 2D class averages, helping to validate the reconstruction.

The refined map was sharpened using phenix.auto_sharpen at the resolution cutoff[57], and a near-atomic resolution model was built de novo into the sharpened map using COOT[58]. We generated a five-layer model using the helical parameters from the final 3D refinement and then refined the structure using phenix.real_space_refine[59]. After the last round of refinement, we adjusted the residue's phi/psi angles to facilitate main chain hydrogen-bonding, and the final model was validated using phenix.comprehensive_validation[60,61]. All the statistics are summarized in Table 1.

**Solvation energy calculation**. The solvation energy was calculated based on work published previously[62]. The solvation energy for each residue was calculated by the sum of the products of the area buried for each atom and the corresponding atomic solvation parameters. The overall energy was calculated by the sum of energies of all residues. Different colors were assigned to each residue, instead of each atom, in the solvation energy map. The energy reported for FUS in Table 2 is the average over 20 NMR models.

**3D environment compatibility search**. 3D environment profiling is performed based on the method described previously[30]. Briefly, an environment category was assigned for the side chains of each residue, and a profile was created using the model structure and its homologous structures based on three features: (1) the secondary structure the residue is in, (2) the fraction of side chain that is covered by polar atoms, and (3) the area buried. A 3D environment profile was created using our mCherry-hnRNPA2-LCD fibril core structure.

**Peptides stabilization energy calculation**. The hnRNPA2 fibril structure is divided into β-sheets composed of six-residue long contiguous segments. Then the surface area buried between all pairwise combinations of these sheets is measured. If the area buried in the dry interface is >80 Å$^2$, the pair of segments is kept as a potential zipper. If the area is <80 Å$^2$, the pair of segments is discarded due to a lack of enough overlap. The energy of the overlapping pairwise segments is measured using our solvation energy calculation described above. The histogram summarizes the energies of 1657 potential zippers extracted from 12 PDB entries of 5 disease-related amyloid protein: amyloid-β (PDB: 5KK3), α-synuclein (PDB: 2N0A, 6CU7, 6CU8), Serum amyloid A (PDB:6DSO), tau (PDB: 5O3L, 5O3T, 6GX5), and TDP-43 (PDB: 6N37, 6N3A, 6N3B, 6N3C).

**Peptide aggregation formation**. Both wildtype and mutant peptide segments are dissolved in water to a final concentration of 10 mg ml$^{-1}$ in 1.5 ml silicon tubes with parafilm seal. Samples were incubated at 25 °C in a Torrey Pine Scientific shaker at level 9 for 90 h and then examined by TEM.

**Segment crystallization**. Crystals of the mutant peptide segment were grown by hanging-drop vapor diffusion. The segment was dissolved in water to a final concentration of 9.69 mg ml$^{-1}$ with 31 mM lithium hydroxide. The reservoir solution contains 0.15 M ammonium acetate, 35% MPD, and 0.1 M Bis–Tris pH 5.5, which is optimized based on the original hit in the commercial kit JCSG 96-well block platescreen from Molecular Dimensions, well H5. Hanging drop contains a 1:1 solution of protein:reservoir.

**X-ray data collection, processing, and structural analysis**. X-ray diffraction data from GNYNVF segment crystals were collected at beamline 24-ID-E of the Advanced Photon Source, Argonne National Laboratory, Argonne, IL, USA, at a wavelength of 0.979 Å and temperature of 100 K. Data were collected using 2.5° oscillations and 150 mm detector distance with an EIGER detector. Indexing and integration of the reflections were done using XDS in space group P2$_1$ and scaled with XSCALE to a resolution of 1.1 Å[63]. The structure was solved by direct method using SHELXD[64]. The atomic-resolution model was manually built and adjusted using COOT[58]. The model is then refined by Refmac[65] with a final $R_{work}/R_{free}$ of 7.5/10.7(%) and 100% of Ramachandran angles favored. The crystal structure image was generated using Pymol. Area buried (Ab) and shape complementarity (Sc) were calculated based on work published previously[66,67]. All the statistics are summarized in Table 3.

**Reporting summary**. Further information on research design is available in the Nature Research Reporting Summary linked to this article.

## Data availability
Atomic coordinates and cryoEM map of the hnRNPA2 LCD fibril were deposited in the Protein Data Bank and Electron Microscopy Data Bank with entry codes: 6WQK and

EMD-21871. The crystal structure GNYNVF was deposited in the Protein Data Bank with entry code: 6WPQ. Other data are available from the corresponding author upon reasonable request.

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

## Acknowledgements
We thank Peng Ge of the Electron Imaging Center for NanoMachines (EICN) for help with cryoEM data collection and processing; We are grateful to Dr. Masato Kato and Dr. Steven L. McKnight from University of Texas, Southwestern Medical Center for sending us the construct of mCherry-hnRNPA2-LCD and their advice on sample preparation and we thank the Whitcome Pre-doctoral Training Program, UCLA Molecular Biology Institute for funding, NSF MCB 1616265, and NIH RF1 AG054022 for support. This work used NE-CAT beamline 24-ID-E (GM124165) and an Eiger detector (OD021527) at the APS (DE-AC02-06CH11357).

## Author contributions
J.L. designed experiments, purified constructs, crystallized peptides, prepared cryo-EM samples, performed X-ray and cryo-EM data collection and processing, and performed data analysis. Q.C. performed cryoEM data processing. D.R.B. assisted in cryoEM data collection and processing. M.P.H. performed LARKS prediction and assisted in model building. M.R.S. performed solvation energy calculation. M.R.S. and D.C. assisted in X-ray data collection and processing and model building. All authors analyzed the results and wrote the manuscript. D.S.E. supervised and guided the project.

## Competing interests
D.S.E. is an advisor and equity shareholder in ADRx, Inc. The other authors declare no competing interests.
