## [Peer Review File · Nature Communications]

REVIEWER COMMENTS

Reviewer #1 (Remarks to the Author):

This is an interesting article on the fibril structure of the low-complexity domain of hnRNPA2 examined by cryoEM and X-Ray crystallography. I was asked to assess the crystal structure. The structure was determined to high/atomic resolution and the validation reports are excellent, indicating the structure is well refined. I also examined sections of the manuscript relevant to the crystal structure, e.g. the results, methods, Figure 4, Table 3, and Supplementary Figure 5. I have some minor comments:

1. There appears to be small differences between the table in the validation report and Table 3. E.g. completeness and I/sigI. Could the authors please double-check?
2. The methods refer to Table 5. Should this be Table 3?
3. Figure 4b would benefit from additional labelling (e.g. of the mating strand and native residues) to help readers cross-reference to the text.
4. In the supplementary figure caption, 'Crystals growth' should be 'Crystal growth'

Reviewer #2 (Remarks to the Author):

Overall it is an interesting study about structural aspects of reversible functional fibrils and beside the structure it discusses biological relevant aspects.

My main issues:

Question 3 at the end of the introduction (line 71) about how a single mutation converts functional to pathogenic fibrils is not clearly answered. At least not based on the presented structures. That makes also part of the title incorrect.

Section "The disease-causing mutant core segment": The hexamer GNYNVF steric zipper crystal structure and the cryo EM structure are really hard to compare. There are too many factors that can influence the structure and it is difficult to draw conclusions about impact of a mutation by comparing the hexameric dimer to a corresponding part in a bigger structure. You would need a structure of the mutant D/V hnRNPA2-LCD fibril to get a better understanding of the impact of the mutation.

The comparison between GNYNVF and GNYNDF with the mutant to be more stable than the WT is an interesting aspect. GNYNVF and GNYNDF dimer structure comparison could also help finding features to explain the different stabilities.

But for now the hexamer structure doesn't seem to be relevant for the discussion and doesn't give a clear answer about the function of the mutation.

Starting at line 240 regarding polymorphism in pathogenic irreversible amyloid fibrils:

A recent ex vivo TTR-fibril reconstruction also showed a monomorphic sample and you can adjust fibrillization conditions to produce monomorphic in vitro fibrils. You can maybe say most studied pathogenic irreversible fibril structures are polymorphic. At the moment it sounds like all of them are polymorphic. If I understand the authors point correctly functional reversible fibrils need to be monomorphic to work, but for pathogenic fibrils it doesn't matter.

The sentence (line 240-242) "In contrast, structures of pathogenic irreversible amyloid identified have shown polymorphism, whether the fibrils were produced ex vivo or in vitro." I think "identified" was meant to be "fibrils"?

Minor stuff:

Supplementary figure 1d a better image to show fibrils at 65° or above would be nice. It's difficult to identify fibrils in the current picture.

I think there is a converting problem for the pdf with the ° sign in the description, I see question

marks in boxes. Also, there are um instead of μm .

Reviewer #3 (Remarks to the Author):

The manuscript by Lu et al. reports the cryoEM structures of fibril formed by hnRNPA2 low-complexity domain (LCD) fused with mCherry. Unlike the pathological amyloid fibrils, it is reversible and displays distinct structural characteristics. Importantly, the Nuclear Localization Signal (NLS) of hnRNPA2 was found in the fibril core, which provide explanation on why Kap β 2 effectively prevents hnRNPA2 fibrilization both in vitro and in cells. In addition, they determined the crystal structure of disease-mutation-containing segment of hnRNPA2, and sought to explain how a point mutation could switch fibril structure from reversible to irreversible form. Overall, the structural data presented to support these findings are generally of high quality. The cryoEM fibril structure of hnRNPA2 LCD presents a novel arrangement of reversible fibril. These structures are exciting and would be of interest not only to the hnRNPA field but also for the amyloid field in general. However, I do have major concerns that need to be addressed and recommended suggestions to improve the work.

Major concerns:

1. It's known that amyloid fibril structures are highly polymorphic. Under different circumstances (e.g. mutation, PTM, co-factors), the same protein may form distinct fibril structures. The authors used the mCherry fused hnRNPA2 LCD to prepare the fibril samples for cryoEM studies. It is questionable whether mCherry which contains over 200 residues alters the overall architecture of the hnRNPA2 LCD fibril. A recent study demonstrates that the florescence protein (GFP) fused to Tau significantly alters the fibril structure of Tau due to the potential steric hindrance (<https://doi.org/10.1101/2020.03.25.998831>). This observation immediately boosts a heated discussion in amyloid field (<https://www.alzforum.org/news/research-news/widely-used-tau-seeding-assay-challenged>).

Therefore, in order to strengthen the physiological relevance of the fibril structure reported here, it's important to rule out the possible influence of the fused mCherry in the formation of hnRNPA2 LCD fibril. The author might need to prepare hnRNPA2 LCD without the mCherry and characterize whether it retains the similar fibril structure. I understand that it will take some time given the coronavirus outbreak. Rather than resolving the atomic fibril structure of hnRNPA2 LCD alone by cryoEM, the authors may use AFM, NS-TEM to compare the fibril structures of the mCherry fused LCD and the LCD alone.

2. The crystal structure of G6F provides very limit yet confusing information in explaining the fibril structures and the reversibility of WT and mutated hnRNPA2 LCD fibril. It's not surprising at all that the backbones of the G6F in the crystal fibril and in cryoEM fibril overlay well, since both of the segments adopt beta-conformation. The author analyzed the dry interface of the G6F homo-zipper from the crystal structure and proposed that it may explain the irreversibility of the mutated fibril. However, to form this homo-zipper interface, the hnRNPA2 LC subunit needs to totally rearrange (due to the steric hindrance) in the protofilament structure. Moreover, the two protofilaments need to further intertwine and bundle together. Thus, if the authors want to include the homo-zipper structure to explain the LC fibril, they need to provide additional structural evidence to support this indeed occurs in the irreversible fibril formed by the mutation rather than in G6F crystal. For instance, the authors may measure the thickness of the irreversible mutation fibrils, at least to demonstrate that it's formed by two protofilament rather than a single one observed in their reversible fibril of hnRNPA2 LC. Otherwise, the authors might need to remove the homo-zipper structure to avoid the confusion and overinterpretation.

Minor comments:

1. To fully assess the reversibility of the hnRNPA2 LCD hydrogel, the authors might need to test

whether the heat-dissolved hnRNPA2 LCD protein can form hydrogel again upon cooling down to 4°C with further incubation.

2. It's very confusing that the authors directly compared the crystallin form of the mutant segment GNYNVF and the fibrillar form of segment GNYNDF, and came up with a conclusion that the mutant segment forms more stable structure than the WT segment. These data can only tell the mutant segment GNYNVF is more prone to crystallize, while the segment GNYNDF tends to form fibril under the condition tested in this study. If the authors want to compare the fibril stability, they need to prepare the fibrillar rather than the crystal form of the mutant segment GNYNVF.

Response: We thank the reviewers for their detailed critiques of our paper and for their helpful suggestions. Accordingly we made the following major changes to the manuscript:

1) We added the Data Availability section with Protein Data Bank and Electron Microscopy Data Bank entry codes for the cryoEM fibril structure and map and the Protein Data Bank code for the peptide crystal structure.

2) We performed energetic calculations in order to address Reviewer #2 and #3's questions about the comparison of the peptide structure to the fibril structure, and added a corresponding section "Implications of the GNYNVF crystal structure for full-length hnRNPA2 LCD D290V fibrils" in the Discussion. We also added the energy histogram as Supplementary Figure 8.

3) To address major concern 2 of Reviewer #3, we added cryoEM images of the wild-type and D290V mutant fibrils as Supplementary Figure 8 and compared the thickness of wild type and mutant fibrils. We added a paragraph in the Discussion explaining the comparison between wild type and mutant fibrils.

Additional changes are explained in the point-by-point.

Reviewer #1 (Remarks to the Author):

This is an interesting article on the fibril structure of the low-complexity domain of hnRNPA2 examined by cryoEM and X-Ray crystallography. I was asked to assess the crystal structure. The structure was determined to high/atomic resolution and the validation reports are excellent, indicating the structure is well refined. I also examined sections of the manuscript relevant to the crystal structure, e.g. the results, methods, Figure 4, Table 3, and Supplementary Figure 5. I have some minor comments:

1. There appears to be small differences between the table in the validation report and Table 3. E.g. completeness and $I/\sigma I$. Could the authors please double-check?

Response: Thank you. A rounding error from two different refinement methods accounts for the difference in completeness. We now report the value in Table 3 as 87%. The value for I/σ calculated by "xtriage" is only an estimation as noted in a footnote of this table. Specifically, phenix's xtriage program estimates intensities from user supplied amplitudes. The I/σ value that we report in Table 3 is correct. It is taken from the XSCALE log file.

2. The methods refer to Table 5. Should this be Table 3?

Response: Thank you for noting this error; we have corrected the reference from Table 5 to Table 3.

3. Figure 4b would benefit from additional labelling (e.g. of the mating strand and native residues) to help readers cross-reference to the text.

Response: Thank you; we have included the labels requested.

4. In the supplementary figure caption, 'Crystals growth' should be 'Crystal growth'

Response: Thank you; we have corrected the caption.

Reviewer #2 (Remarks to the Author):

Overall it is an interesting study about structural aspects of reversible functional fibrils and beside the structure it discusses biological relevant aspects.

My main issues:

Question 3 at the end of the introduction (line 71) about how a single mutation converts functional to pathogenic fibrils is not clearly answered. At least not based on the presented structures. That makes also part of the title incorrect.

Section “The disease-causing mutant core segment”: The hexamer GNYNVF steric zipper crystal structure and the cryoEM structure are really hard to compare. There are too many factors that can influence the structure and it is difficult to draw conclusions about impact of a mutation by comparing the hexameric dimer to a corresponding part in a bigger structure. You would need a structure of the mutant D/V hnRNPA2-LCD fibril to get a better understanding of the impact of the mutation.

The comparison between GNYNVF and GNYNDF with the mutant to be more stable than the WT is an interesting aspect. GNYNVF and GNYNDF dimer structure comparison could also help finding features to explain the different stabilities.

But for now the hexamer structure doesn't seem be relevant for the discussion and doesn't give a clear answer about the function of the mutation.

Response: Thank you. We agree with reviewer #2 that we cannot accurately predict the effect of a single-mutation on the fibril structure of full length hnRNPA2 knowing only the mutation's affect on a 6-residue segment of hnRNPA2. However, this hexapeptide structure gives us important clues. (1) As shown in the manuscript, the mutant hexamer GNYNVF is more prone to form crystals than the wildtype GNYNDF, as indicated by its greater energetic stability. (2) This mutant zipper's stabilization energy is more favorable than any of the other 98 hexamer segments that constitute the fibril. As the energy histogram below (also Supplementary Figure 8a) suggests, the stabilization energy of this mutant zipper GNYNVF is more akin in stability with pathogenic amyloid than the weaker zippers of our cryoEM structure. Based on this energetic analysis, we hypothesize that when this mutation is formed, the mutant zipper can drive the assembly of a more stable fibril form. (3) If the GNYNDF segment of full-length hnRNPA2 assembles as it does in the peptide crystal structure, the remaining residues would necessarily adopt a different conformation than the wild type fibril in order to avoid steric clash with the hexapeptide assembly. The purpose of Figure 4b is simply to show this incompatibility between the wild type fibril and a hypothetical mutant assembled with the same geometry as our well-determined crystal structure. The incompatibility illustrated in this superposition suggests that the wild type and mutant hnRNPA2 fibrils probably have very different core structures. This hypothesis is speculative, but it is consistent with the data we have.

In the revised Discussion of the manuscript, we added a section “Implications of the

GNYNVF crystal structure for full-length hnRNPA2 LCD D290V fibrils” to address reviewer #2’s concern, as follows: “We offer a speculative hypothesis of how a single-point missense mutation can alter the fibril structure. This hypothesis is consistent with the data we have. As indicated by its greater stabilization energy, the mutant segment GNYNVF is more prone to form crystals than the wild type segment GNYNDF (Figure 4). In addition, we calculated the stabilization energy of the mutant segment and all hexamer segments that constitute the wild type hnRNPA2 LCD fibril, FUS LCD fibril, and pathogenic amyloids. Unsurprisingly, the mutant segment is an outlier in terms of having greater energetic stability than any hexameric segment in either hnRNPA2 or FUS (Supplementary Figure 8a). This energetic analysis supports our hypothesis that when this mutation forms, the mutant segment can drive the assembly of a more stable fibril.”

We have included the energy analysis as Supplementary Figure 8a.

We also added a section “Peptides stabilization energy calculation” in Methods explaining how we calculate the peptide stabilization energy, as follows: “HnRNPA2 fibril structure is divided into a series of overlapping 6-residue long contiguous segments. Then the surface area buried between all pairwise segments of these sheets is measured. If the area buried in the dry interface is greater than 80 Å², the pair of segments is kept as a potential zipper. If the area is less than 80 Å², the pair of segments is discarded due to a lack of enough overlap. The energy of the overlapping pairwise segments is measured using our solvation energy calculation described above. The pathogenic amyloids we used here for analysis include: amyloid-β (PDB: 5KK3), α-synuclein (PDB: 2N0A, 6CU7, 6CU8), AA amyloid (PDB:6DSO), tau (PDB: 5O3L, 5O3T, 6GX5), and TDP43 (PDB: 6N37, 6N3A, 6N3B, 6N3C).”

Starting at line 240 regarding polymorphism in pathogenic irreversible amyloid fibrils:

A recent ex vivo TTR-fibril reconstruction also showed a monomorphic sample and you can adjust fibrillization conditions to produce monomorphic in vitro fibrils. You can maybe say most studied pathogenic irreversible fibril structures are polymorphic. At the moment it sounds like all of them are polymorphic. If I understand the authors point correctly

functional reversible fibrils need to be monomorphic to work, but for pathogenic fibrils it doesn't matter.

Response: Thank you; we clarified the sentence in the Discussion, paragraph 3 in the section "Common features of LCDs of functional, amyloid-like fibrils: hnRNPA2 versus FUS", as follows: "In contrast, the majority of pathogenic, irreversible amyloid fibril structures have shown polymorphism". That is, we no longer claim all pathogenic amyloid fibrils are polymorphic.

The sentence (line 240-242) "In contrast, structures of pathogenic irreversible amyloid identified have shown polymorphism, whether the fibrils were produced ex vivo or in vitro." I think "identified" was meant to be "fibrils"?

Response: Thank you; we clarified the sentence in the Discussion, paragraph 3 in the section "Common features of LCDs of functional, amyloid-like fibrils: hnRNPA2 versus FUS", as follows: "In contrast, the majority of pathogenic, irreversible amyloid fibril structures have shown polymorphism".

Minor stuff:

Supplementary figure 1d a better image to show fibrils at 65° or above would be nice. It's difficult to identify fibrils in the current picture.

Response: Thank you; we replaced figure 1d with a clearer image.

I think there is a converting problem for the pdf with the ° sign in the description, I see question marks in boxes. Also, there are um instead of μm.

Response: Thank you; we corrected "um" to "μm" throughout the manuscript. We will make sure we fix the converting problem when resubmitting.

Reviewer #3 (Remarks to the Author):

The manuscript by Lu et al. reports the cryoEM structures of fibril formed by hnRNPA2 low-complexity domain (LCD) fused with mCherry. Unlike the pathological amyloid fibrils, it is reversible and displays distinct structural characteristics. Importantly, the Nuclear Localization Signal (NLS) of hnRNPA2 was found in the fibril core, which provide explanation on why Kapβ2 effectively prevents hnRNPA2 fibrilization both in vitro and in cells. In addition, they determined the crystal structure of disease-mutation-containing segment of hnRNPA2, and sought to explain how a point mutation could switch fibril structure from reversible to irreversible form. Overall, the structural data presented to support these findings are generally of high quality. The cryoEM fibril structure of hnRNPA2 LCD presents a novel arrangement of reversible fibril. These structures are exciting and would be of interest not only to the hnRNPA field but also for the amyloid field in general. However, I do have major concerns that need to be addressed and recommended suggestions to improve the work.

Major concerns:

1. It's known that amyloid fibril structures are highly polymorphic. Under different circumstances (e.g. mutation, PTM, co-factors), the same protein may form distinct fibril structures. The authors used the mCherry fused hnRNPA2 LCD to prepare the fibril samples for cryoEM studies. It is questionable whether mCherry which contains over 200 residues alters the overall architecture of the hnRNPA2 LCD fibril. A recent study demonstrates that the fluorescence protein (GFP) fused to Tau significantly alters the fibril structure of Tau due to the potential steric hindrance (<https://doi.org/10.1101/2020.03.25.998831>). This observation immediately boosts a heated discussion in amyloid field (<https://www.alzforum.org/news/research-news/widely-used-tau-seeding-assay-challenge>).

Therefore, in order to strengthen the physiological relevance of the fibril structure reported here, it's important to rule out the possible influence of the fused mCherry in the formation of hnRNPA2 LCD fibril. The author might need to prepare hnRNPA2 LCD without the mCherry and characterize whether it retains the similar fibril structure. I understand that it will take some time given the coronavirus outbreak. Rather than resolving the atomic fibril structure of hnRNPA2 LCD alone by cryoEM, the authors may use AFM, NS-TEM to compare the fibril structures of the mCherry fused LCD and the LCD alone.

Response: Thank you for raising this important point, which we believe we have reasonably addressed. We have attempted to characterize hnRNPA2 LCD without mCherry. We found that hnRNPA2 LCD is highly insoluble without mCherry and tends to aggregate immediately instead of forming fibrils. The paper by Kaniyappan et al. (GFP-Tau paper reviewer #3 suggested) has shown that when GFP **is close to the protein core** (which is the repeat domain of Tau in the paper), protein fibrillation is inhibited, thus affecting the core structure, but full-length Tau is not as affected by the GFP tag as is the repeat domain because of a long linker ("fuzzy coat") between the GFP tag and the protein core. Similarly, as indicated in the discussion of our manuscript, in between the mCherry tag and hnRNPA2 LCD core, there is an 86-amino-acid long peptide which is similar to the "fuzzy coat" in the GFP-Tau paper. Moreover, hnRNPA2, as a functional protein, should be honed evolutionarily to be monomorphic to function biologically. Thus we propose that mCherry would not be expected to significantly alter the structure of hnRNPA2 LCD.

To raise this point for readers and to clarify our argument, we include a Discussion subsection "Effects of the surrounding mCherry-tag on the structure of mC-hnRNPA2-LCD fibrils" in the manuscript that specifically raises and addresses this concern. We also added further arguments in this section. Specifically, in the first paragraph of this section, we inserted "We actively sought to characterize the hnRNPA2 LCD fibril structure without the mCherry tag; however, we found that hnRNPA2 LCD alone is highly insoluble and tends to aggregate. We thus included the mCherry tag in our experiments. mCherry is visible during cryoEM processing.". We added a second reason: "When the fluorescent tag is sufficiently separated from the fibril core, the tag can freely fold around the fibril core as a "fuzzy coat", so that fibrillization and core packing are not disturbed". At the end of this paragraph, we added a third reason: "Third, the mCherry tag "fuzzy coat" is not visible in the cryoEM 3D classification (Supplementary figure 9d), suggesting there are no disruptive interactions between the large mCherry tag and the hnRNPA2 LCD fibril core.". In short, we believe we have clearly articulated in the text the concern raised by Reviewer #3, and set forth reasons

indicating that the mCherry tag does not distort the fibril structure.

And because we have added in the peptide energy figure in the preceding section, we changed the original Supplementary Figure 8 to Supplementary Figure 9.

2. The crystal structure of G6F provides very limited yet confusing information in explaining the fibril structures and the reversibility of WT and mutated hnRNPA2 LCD fibril. It's not surprising at all that the backbones of the G6F in the crystal fibril and in cryoEM fibril overlay well, since both of the segments adopt beta-conformation. The author analyzed the dry interface of the G6F homo-zipper from the crystal structure and proposed that it may explain the irreversibility of the mutated fibril. However, to form this homo-zipper interface, the hnRNPA2 LC subunit needs to totally rearrange (due to the steric hindrance) in the protofilament structure. Moreover, the two protofilaments need to further intertwine and bundle together. Thus, if the authors want to include the homo-zipper structure to explain the LC fibril, they need to provide additional structural evidence to support this indeed occurs in the irreversible fibril formed by the mutation rather than in G6F crystal.

For instance, the authors may measure the thickness of the irreversible mutation fibrils, at least to demonstrate that it's formed by two protofilament rather than a single one observed in their reversible fibril of hnRNPA2 LC. Otherwise, the authors might need to remove the homo-zipper structure to avoid the confusion and overinterpretation.

Response: Thank you for bringing up this very good question. This question was also raised by Reviewer #2. In addition to our answer to Reviewer #2, we have compared the cryoEM images of wild type and mutant fibrils. The images show that mutant fibrils are more heterogeneous and thicker than the wild type fibrils. The greater thickness supports our hypothesis that the D290V mutation causes the segment GNYNVF to form a homo-zipper and drive the assembly of hnRNPA2 LCD fibrils. However, high-resolution structure determination of the mutant fibrils is needed to confirm our hypothesis. We are actively seeking the mutant fibril structure and will determine it in the future. We believe this work constitutes a separate future study.

Nevertheless, to raise and clarify this point, we have added a paragraph in the Discussion section "Implications of the GNYNVF crystal structure for full-length hnRNPA2 LCD D290V fibrils" and the cryoEM comparison images as Supplementary Figure 8c&d. The addition is as follows: "If the wild type segment GNYNDF of full-length hnRNPA2 assembles as it does in the peptide crystal structure, the remaining residues would necessarily adopt a different conformation than the wild type fibril in order to avoid steric clash with the hexapeptide assembly. Our cryoEM images offer evidence that the mC-hnRNPA2-D290V-LCD fibrils are more heterogeneous and thicker than the wild type mC-hnRNPA2-LCD fibrils (Supplementary Figure 8c&d). This supports our hypothesis that the D290V mutation drives the mutant segment GNYNVF to adopt a homo-zipper conformation and drive the fibril assembly into a different structure with two protofilaments."

Minor comments:

1. To fully assess the reversibility of the hnRNPA2 LCD hydrogel, the authors might need to test whether the heat-dissolved hnRNPA2 LCD protein can form hydrogel again upon cooling down to 4°C with further incubation.

Response: Thank you for raising this point, which we now address in a new paragraph in the first section of the Discussion: “Heated mC-hnRNPA2-LCD, instead of remaining as a fibril network, is broken into oligomer-like materials, and does not reform a hydrogel upon cooling. We hypothesize that in this *in vitro* experiment, the environment lacks biological factors such as chaperones to achieve reversibility. Nevertheless, the smaller stabilization solvation energy compared to pathogenic amyloid indicates the reversibility of hnRNPA2 LCD fibrils (Figure 3).”

2. It’s very confusing that the authors directly compared the crystallin form of the mutant segment GNYNVF and the fibrillar form of segment GNYNDF, and came up with a conclusion that the mutant segment forms more stable structure than the WT segment. These data can only tell the mutant segment GNYNVF is more prone to crystallize, while the segment GNYNDF tends to form fibril under the condition tested in this study. If the authors want to compare the fibril stability, they need to prepare the fibrillar rather than the crystal form of the mutant segment GNYNVF.

Response: Thank you for raising this point as did Reviewer #2, in main issue 1. Please see our response to Reviewer #2, which we hope clarifies the point for readers.